# Towards Chapter-to-Chapter Literary Translation via Large Language Models

## Abstract

Discourse phenomena in existing document-level translation datasets are sparse, which has been a fundamental obstacle in the development of context-aware machine translation models. Moreover, most existing document-level corpora and context-aware machine translation methods rely on an unrealistic assumption on sentence-level alignments. To mitigate these issues, we first curate a novel dataset of Chinese-English literature, which consists of 132 books with intricate discourse structures. Then, we propose a more pragmatic and challenging setting for context-aware translation, termed chapter-to-chapter (Ch2Ch) translation, and investigate the performance of commonly-used machine translation models under this setting. Furthermore, we introduce a potential approach of finetuning large language models (LLMs) within the domain of Ch2Ch literary translation, yielding impressive improvements over baselines. Through our comprehensive analysis, we unveil that literary translation under the Ch2Ch setting is challenging in nature, with respect to both model learning methods and translation decoding algorithms.

## 1 Introduction

Despite the efforts on developing context-aware machine learning systems to meaningfully exploit inter-sentential information, recent work has investigated the fundamental obstacles in existing document-level translation datasets and context-aware machine translation models (Jin et al., 2023). First, existing datasets lack the necessary contextual information and/or discourse phenomena for meaningful document-level translation (Lupo et al., 2022). Second, existing predominant context-aware translation methods assume that sentence-level alignments are available during training, which does not accurately represent real-world translation scenarios (Thai et al., 2022; Jin et al., 2023).

To remedy the issues, recent work has pivoted to literary translation and proposed a more realistic paragraph-to-paragraph setting, given that literary texts typically contain complex discourse structures that mandate a document-level frame of reference. Thai et al. (2022) released Par3, a paragraph-level translation dataset sourced from recently-published 118 novels in 19 languages (about 6 novels per language on average). Jin et al. (2023) curated Para2Para, a small-scale dataset consisting of 10,545 parallel paragraphs across six novels. However, these datasets are either in small scale or the reference translations are automatically generated from machine translation systems (e.g. Google Translate (Wu et al., 2016) and fine-tuned GPT-3 (Brown et al., 2020)). In addition, there still exist some serious limitations in the paragraph-to-paragraph translation setting, including limited contextual information and equivocal paragraph splits in literary texts.

Large language models (LLMs) with decoder-only Transformer architectures have demonstrated outstanding performance as sentence-level translation systems (Vilar et al., 2023; Jiao et al., 2023; Kocmi & Federmann, 2023; Zhang et al., 2023; Yang et al., 2023). In the aspect of context-aware translation, recent studies have employed decoder-only LLMs to translate entire paragraphs using few-shot in-context learning methods, yielding impressive translation quality (Karpinska & Iyyer, 2023). However, how to finetune LLMs to process context-aware translation for literary texts in a more realistic and challenging scenario remains under-explored.

In this paper, we propose a more pragmatic and challenging setting for context-aware translation, named *chapter-to-chapter* (Ch2Ch), associated with a carefully curated dataset of Chinese-English literature. The dataset consists of 132 literary books, together with professional translations in Chinese. Then we investigate the performance of commonly-used machine translation models under

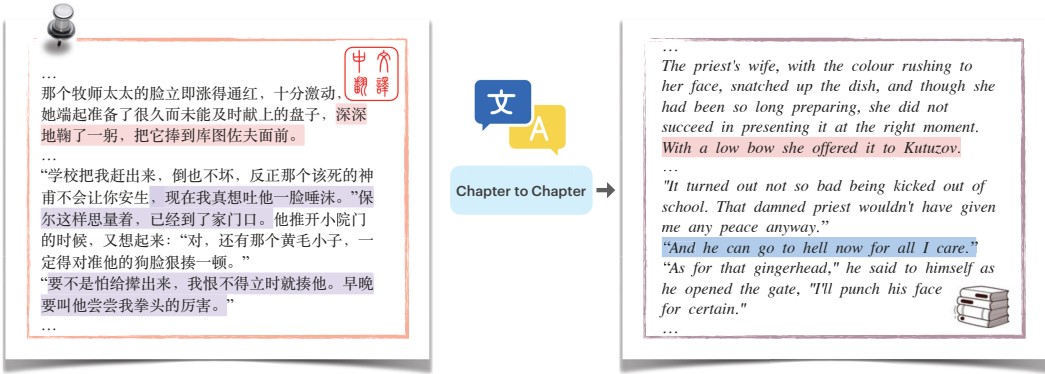

Figure 1: An example of of CH2CH translation. Sentence Misalignment: Red parts are where a source sentence is separated into multiple sentences in the corresponding translation; blue parts are added by translators without a corresponding source segment; violet parts are deleted by translators.

the proposed setting and dataset. In addition, we investigate the efficacy of applying LLMs in context-aware chapter-to-chapter literary translation and highlight several key challenges that impede the progress. Our main contributions are outlined as follows:

- We propose a more realistic setting for literary translation: chapter-to-chapter(CH2CH) translation, wherein a document is translated at the granularity of chapters. To support it, we release a chapter-aligned Chinese-English dataset (JAM), comprising 4,194 parallel chapters extracted from 132 novels, to catalyze future research endeavors.
- Through comprehensive analysis, we unveil the challenges in chapter-level translation, including long-context model training and decoding strategies.
- With empirical experiments, we evaluate the performance of recent trending LLMs on the JAM dataset and propose an effective fine-tuning procedure tailored for LLMs to generate coherent translations of literary novels.

## 2 PRELIMINARY BACKGROUND

### 2.1 CONTEXT-AWARE NEURAL MACHINE TRANSLATION

**Sentence-aligned Translation** In the sentence-aligned setting of context-aware machine translation, we assume that the source and target sentences in a parallel document are well-aligned. Formally, given a document $D$ comprising a set of source sentences $\boldsymbol{X} = \{\boldsymbol{x}_1, \boldsymbol{x}_2, ..., \boldsymbol{x}_d\}$, there are the same number of sentences $\boldsymbol{Y} = \{\boldsymbol{y}_1, \boldsymbol{y}_2, ..., \boldsymbol{y}_d\}$ in the target side, which are aligned with sentences in $\boldsymbol{X}$ by the indices. The context-aware neural machine translation (NMT) model computes the probability of translating the source sentence $\boldsymbol{x}_i$ conditioned on the context $C_i$, wherein $0 \leq i \leq d$:

$$P_{\text{SentAlign}}(\boldsymbol{y}_i|\boldsymbol{x}_i, \boldsymbol{C}_i, \theta) = \prod_{j=1}^{N} P(y_i^j|y_i^{<j}, \boldsymbol{x}_i, C_i; \theta). \tag{1}$$

where $C_i$ are contextual sentences surrounding $\boldsymbol{x}_i$ and/or $\boldsymbol{y}_i$. As illustrated in Figure 1, sentence-aligned translation does not accurately represent real-world translation scenarios.

**Paragraph-to-Paragraph Translation** To get rid of the assumption of sentence-level alignments and leverage richer contextual information, recent work (Thai et al., 2022; Jin et al., 2023) proposed a paradigm shift towards paragraph-to-paragraph (PARA2PARA) translation to relax the alignment assumption from sentence-level to paragraph-level. Concretely, a document $D$ contains a set of aligned parallel paragraphs, $\boldsymbol{X} = \{\boldsymbol{X}_1, \boldsymbol{X}_2, ..., \boldsymbol{X}_d\}$ and $\boldsymbol{Y} = \{\boldsymbol{Y}_1, \boldsymbol{Y}_2, ..., \boldsymbol{Y}_d\}$. Each pair of aligned paragraphs $\boldsymbol{X}_i$ and $\boldsymbol{Y}_i$ do not necessarily contain the same number of sentences:

$$P_{\text{Para2Para}}(\boldsymbol{Y}_i|\boldsymbol{X}_i, \theta) = \prod_{j=1}^{N} P(Y_i^j|Y_i^{<j}, \boldsymbol{X}_i; \theta) \tag{2}$$

where $Y_i^{<j}$ are all previously translated tokens in a paragraph. However, in literary texts the splits of paragraphs are equivocal, which limits the application of PARA2PARA to real-world scenario.

## 2.2 Datasets

Most commonly used corpora, including IWSLT-17 (Cettolo et al., 2012), NewsCom (Tiedemann, 2012), Europarl (Koehn, 2005), and OpenSubtitles (Lison et al., 2018) are sourced from news articles or parliamentary proceedings. Until recently, some document-level parallel corpora of literary texts have been released. Jiang et al. (2023) curated Bilingual Web Books (BWB), a sentence-aligned corpus that retains document-level information. BWB contains 9.6 million sentence pairs sourced from Chinese web novels and their corresponding English translations. However, BWB still follows the sentence-level alignment constrains. To support PARA2PARA translation, Thai et al. (2022) introduced PAR3, a paragraph-aligned corpus obtained through both human and automatic translators, containing multilingual non-English novels and their English translations. Another paragraph-aligned corpus, introduced by Al Ghussin et al. (2023), consists of parallel paragraphs extracted from Paracrawl (Bañón et al., 2020) using automatic sentence alignments. This corpus includes data crawled from the Internet spanning various domains.

## 2.3 Translation with Large Language Models

LLMs are not explicitly trained on parallel data for translation, yet they possess a profound understanding of languages and can produce coherent text, serving as a valuable foundation for translation tasks (Li et al., 2024). Particularly for resource-rich languages, colossal models with decoder-only architecture, such as GPT-4 (OpenAI et al., 2024), have approached or even exceeded traditional encoder-decoder models on sentence-level benchmarks and can generate more coherent and human-like translations drawing upon their extensive comprehension of both languages (Robinson et al., 2023; Hendy et al., 2023). Xu et al. (2023a) proposed a two-stage procedure to finetune Llama2-7b (Touvron et al., 2023) with a small amount of sentence-level parallel data and obtained impressive improvements over standard sentence-level NMT baselines without LLMs.

# 3 JAM: Chapter-Aligned Literary Translation Dataset

## 3.1 Chapter-to-Chapter Translation

In literary texts, the lengths of paragraphs vary and the splits of paragraphs are equivocal, particularly when dialogues are involved. For instance, in novels, dialogue lines are often presented as separate paragraphs, making it challenging to ensure accurate translations without access to the preceding context. As illustrated by the two examples shown in Table 1, there are instances where multiple paragraphs from the source side are merged into one paragraph on the target side, and vice versa.

To address this issue, we propose *chapter-to-chapter* (CH2CH) translation, a pragmatic and challenging setting, by extending context-aware translation to chapter-level. Comparing to paragraph-level alignments, chapter-level alignments provide the model with more comprehensive context from both the source and target texts. This richer context theoretically offers greater potential for improvements and helps mitigate issues such as tense mismatches, particularly in languages like Chinese that lack explicit tense markers (Sun et al., 2020).

To conduct experiments and facilitate future research endeavours on CH2CH translation, we curate a chapter-aligned dataset of English-Chinese literature, named JAM, which comprises 132 English classic novels alongside professional Chinese translations. In professional literary translation, translators often leverage contexts to enhance the fluency and readability of the translation. To this end, translations may not strictly adhere to sentence alignment[1], and some typical sentence misalignment types are listed below, an example is shown in Figure 1 illustrates:

**INSERT** : new sentence(s) is added by translators and does not have a corresponding source segment.

**DELETE** : a source sentence(s) is deleted by translators in translation.

**SPLIT** : a source sentence is separated into multiple sentences in the corresponding translation.

As such, chapter-to-chapter(CH2CH) translation is challenging in nature, given that chapters typically are lengthy and contain complex discourse structure. Detailed experimental results and analysis are provided in Section 5.1.

---

[1] In 50 sampled paragraphs from JAM there are 18 paragraphs with sentence mis-alignments.

| Source | Target |
|---|---|
| "To think what we have been brought to!" Kutuzov cried suddenly, in a voice full of feeling, Prince Andrey's story evidently bringing vividly before him the position of Russia. | "弄到什么地步……到什么地步！"库图佐夫突然说，他声音激动，显然，从安德烈公爵的叙述中，他清楚地想象到俄国目前的处境。"给我一段时间，给我一段时间！"他脸上带着愤怒的表情又说，很明显，他不愿继续这个使他激动的话题，他说："我叫你来，是想让你留在我身边。" |
| "Wait a bit; wait a bit!" he added, with a vindictive look in his face, and apparently unwilling to continue a conversation that stirred him too deeply, he said: | |
| "I sent for you to keep you with me." | |
| "We must, if everyone wants to; there is no help for it … But, mark my words, my dear boy! The strongest of all warriors are these two—time and patience. They do it all, and our wise counsellors n'entendent pas de cette oreille, voilà le mal. Some say ay, and some say no. What's one to do?" he asked, evidently expecting a reply. "Come, what would you have me do?" he repeated, and his eyes twinkled with a profound, shrewd expression. "I'll tell you what to do," he said, since Prince Andrey did not answer. "I'll tell you what to do. Dans le doute, mon cher"—he paused—"abstiens-toi." He articulated deliberately the French saying. | "打一仗是可以的，如果大家都愿意的话，没有什么可说的……可是要知道，亲爱的朋友：没有比忍耐和时间这两个战士更强的了，这两位什么都能办成。可是顾问们不肯听这个，困难就在这里。一些人要这样，另一些又不这样。怎么办呢？"他问，显然在等着回答。

"你说说看，我怎么办？"他重复着，眼睛显得深沉、睿智。

"我告诉你怎么办。如果你犹豫不决，亲爱的，"他停了一下，"那你先干别的。"他慢条斯理地一字一句地说。 |

Table 1: Examples of paragraph misalignment. Each line represents an individual paragraph in the original text.

## 3.2 DATA CONSTRUCTION AND QUALITY CONTROL

We collect 132 bilingual literary books across different genres from the Internet, and format data by manually correcting chapter-level alignment[2]. Subsequently, we perform standard data cleaning steps (e.g. punctuation normalization) and filter the chapter pairs with a sequence length ratio $> 3.0$. The refined dataset contains a total of 4194 aligned chapters. The statistics of this dataset are shown in Table 2 [3], and detailed

|  | CHAP. # | SENTENCE # (EN/ZH) | WORD # (EN/ZH) |
|---|---|---|---|
| TRAIN | 3546 | 334.8K / 445.0K | 7.4M / 8.6M |
| VALID | 391 | 36.5K / 47.9K | 796.1K / 935.9K |
| TEST | 257 | 29.5K / 40.6K | 648.4K / 795.3K |
| TOTAL | 4194 | 400.7K / 533.6K | 8.8M / 10.4M |

Table 2: JAM Corpus Statistics.

corpus information is in Appendix A.1. The dataset is split into train, valid, and test sets. We randomly select 16 books as the test set. The remaining corpus of 3937 chapters from 116 books was then split into an 90% training set and a 10% validation set.

## 4 EXPERIMENTAL SETUP

### 4.1 BASELINES

To examine the inherent capacity of the model in the translation task, we perform a benchmarking analysis against two baseline categories:

**Encoder-Decoder Architecture** We use the Transformer (Vaswani et al., 2017) `base` version, which consists of 6 encoder layers, 6 decoder layers, a model dimension of 512, and an FFN hidden dimension of 2048.

**Decoder-only Architecture** Compared to the prevalent encoder-decoder architecture, the decoder-only framework is often simpler in architecture and computationally efficient (Fu et al., 2023). In the CH2CH translation task, we train the decoder-only model by concatenating each source chapter with its corresponding target chapter, demarcated by a `<SEP>` token, and ended with an `<EOS>` token:

```
<SRC Chapter> <SEP> <TGT Chapter> <EOS>
```

The model architecture is shown in Figure 2.

Motivated by Zhang et al. (2018), we experiment with training a baseline model on the JAM dataset from scratch, as well as incorporating pre-trained baselines. In the pre-trained baselines, the model is first trained on the sentence-level WMT22 Zh→En dataset (Kocmi et al., 2022), before further fine-tuning on the JAM dataset.

---

[2]We select literary works with chapter breaks, then manually check the alignments of the first and last paragraphs for each chapter.

[3]English sentences are split by white space; Chinese sentences are segmented using the Jieba package.

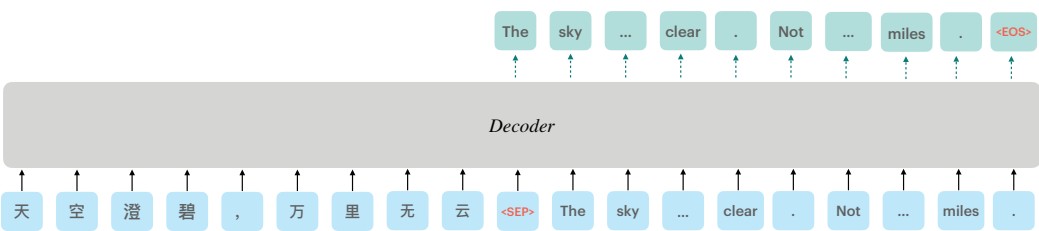

Figure 2: Decoder-only architecture.

**Zero-shot Evaluation**   Recent work has showcased the proficiency of LLMs in sentence-level translation. To further probe the ability of LLMs in translating literary, we randomly sample 63 chapters from JAM test set and conduct a zero-shot evaluation on the sampled instances to compare with the following models:

```
Prompt
Translate this from [src lang] to [tgt lang]:
[src lang]: <src chapter>
[tgt lang]:
```

Figure 3: Prompt template for LLMs.

NLLB-200-3.3B (Team et al., 2022): an encoder-decoder LLM, with 3.3b parameters.

LLAMA2-7B (Touvron et al., 2023): a generative text model with 7b parameters.

ALMA-7B (Xu et al., 2023a): finetuned on 5 language pairs from Llama2-7b for translation.

GPT-4 (OpenAI et al., 2024): a pre-trained large-scale multi-modal model.

Building upon Xu et al. (2023a), we prepend a fixed prompt (Figure 3) to each chapter.

**Finetuning**   We select ALMA-7B to finetune on JAM because of its impressive gains in translation tasks compared to other LLMs; its fine-tuning process is divided into two phrases: first, ALMA-7B-Stage1 finetuned LLAMA2-7B exclusively on monolingual data; then, the second stage ALMA-7B-Stage2 is subsequently finetuned on parallel data. Specifically, we finetune ALMA-7B-Stage1 on JAM to investigate whether pretraining with sentence-level parallel data is beneficial prior to fine-tuning on chapter-level data. We use causal language modeling (CLM) loss for finetuning and restrict loss computation only to the target tokens.

### 4.2 HANDLING LONG CHAPTERS IN TRAINING AND DECODING

As some chapters exceed the maximal context length of some models, we equally segment those chapters into chunks, ensuring that each chunk contains less than 2048 tokens in both Zh and En sides. Data and pre-processing details are in Appendix B.1.

During decoding, we also pack the maximum number of sentences into blocks within 2048 tokens. The model does not know how many sentences to generate in advance and decoding stops when <EOS> is predicted. As illustrated in Figure 2, <EOS> in our experiments is used to indicate the end of translation, not the end of a sentence.

### 4.3 EVALUATION

For all tasks, we report both sentence-level (e.g., BLEU (Papineni et al., 2002), METEOR (Banerjee & Lavie, 2005) and COMET (Rei et al., 2020)) and document-level automatic metrics in evaluation. In particular, we analyze the translation quality of LLMs related to specific discourse phenomena such as pronoun ellipsis, named entity coreference by BlonDe score (Jiang et al., 2022).

## 5 EXPERIMENTAL RESULT AND ANALYSIS

### 5.1 CHAPTER-TO-CHAPTER MACHINE TRANSLATION TASK IS CHALLENGING IN NATURE.

Motivated by Zhang et al. (2018), we experiment with training a baseline model on the JAM dataset from scratch, as well as incorporating a two-stage training procedure, in which the model is first trained on the sentence-level WMT22 Zh→En dataset (Kocmi et al., 2022), before further fine-tuning on the JAM dataset.

| Model | WMT | JAM | BLEU | BlonDe | | | | | COMET |
|---|---|---|---|---|---|---|---|---|---|
| | | | | all | pron. | entity | tense | d.m. | |
| Encoder-Decoder | ✗ | ✓ | 1.87 | 8.70 | 49.23 | 19.22 | 42.30 | 17.21 | 0.4128 |
| Decoder-only | ✗ | ✓ | 1.09 | 7.23 | 47.46 | 20.77 | 40.40 | 16.54 | 0.4187 |
| Encoder-Decoder | ✓ | ✓ | 14.38 | 31.08 | **89.78** | 11.36 | **86.88** | **81.96** | 0.6617 |
| Decoder-only | ✓ | ✓ | 13.35 | 30.06 | 84.28 | 14.59 | 80.23 | 76.81 | 0.6377 |
| ALMA-7B-Stage1 | ✗ | ✓ | 15.70 | 33.46 | 74.28 | 30.62 | 70.11 | 71.72 | 0.7806 |
| ALMA-7B-Stage2 | ✓ | ✓ | **18.80** | **36.90** | 81.34 | **32.72** | 77.83 | 76.81 | **0.8025** |

Table 3: Automatic metric results on JAM test set. Note here chapters are segmented by maximum 2048 tokens. ALMA-7B-Stage1 is only fine-tuned on monolingual data. ALMA-7B-Stage2 fine-tunes ALMA-7B-Stage1 on high-quality parallel data. (✗) denotes no fine-tuning on corresponding dataset; (✓) denotes fine-tuning. **Bold** denotes best performance.

As illustrates in Table 3, Encoder-Decoder and Decoder-only Transformer models trained from scratch on JAM significantly under-perform the models trained with the 2-stage procedure. The significant performance gap demonstrates the challenging nature of CH2CH (e.g., 1.87 and 1.09 on BLEU), i.e., the inherent difficulty of training on chapter-level, long-sequence data. Translation models that trained with the 2-stage procedure to leverage the sentence-level WMT22 exhibit a notable improvement, attesting the difficulty of the CH2CH translation task.

## 5.2 EFFECTIVE FINE-TUNING AND DECODING STRATEGY

**Does sentence-level fine-tuning help?** We next investigate the prerequisite of sentence-level fine-tuning prior to the training on JAM dataset by comparing ALMA-7B-Stage1 and ALMA-7B-Stage2 respectively, with the latter has been fine-tuned on sentence-level parallel datasets. Table 3 indicates that such sentence-level fine-tuning improves BLEU from 15.7 to 18.80 and BlonDe from 33.46 to 36.95, suggesting that fine-tuning at sentence-level contributes positively to the accuracy of chapter-level literary translation.

In contrast, the improvement on COMET is marginal, possibly attributable to COMET's focus on assessing the coherence and fluency of the generated translations. These qualities might already be sufficiently robust in an LLM.

**Repetition Problem in Decoding.** Deutsch et al. (2023) founds that translation does not degrade as the sequence becomes longer. However, according to our results, this is not universally the case; the effectiveness of translation diminishes as the context becomes really lengthy. To investigate the insights, we examine the translations of JAM test set on the fine-tuned ALMA-7B-Stage2 model and observe a notable pattern of undesirable repetitions—either phrases or entire sentences—emerges within the translations.

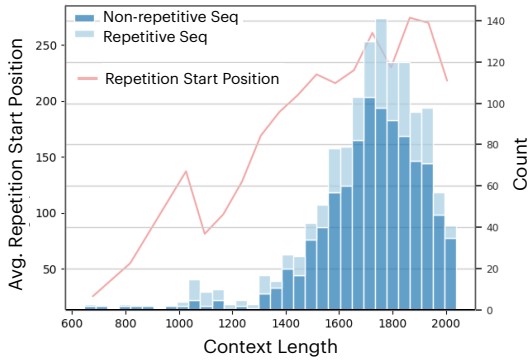

Figure 4: Repetition distribution.

Specifically, 26.4% of the translations within our test set exhibit some form of repetition. As illustrates in Figure 4, repetition occurs predominantly located within the first half of the translations (Shown as the red curve). Furthermore, sentences exceeding 1300 tokens are more likely to generate repetitive words, phrases or sentences[4]. This observation is consistent with earlier studies indicating text generation with LLMs often results in consecutive sentence-level repetitions, attributed to the use of maximization-based decoding algorithms.(Holtzman et al., 2020; Xu et al., 2023b). The detailed analysis by Xu et al. (2022) sheds light on the underlying causes: these models have an inherent tendency to repeat previous sentences, and they tend to overestimate the probability of repeated sequences. This repetition problem is particularly evident in long-context translation, where increasing the chunk length amplifies the risk of the model falling into repetitive loops.

[4]Repetition analysis for all zero-shot generations across various architectures are in Appendix B.4

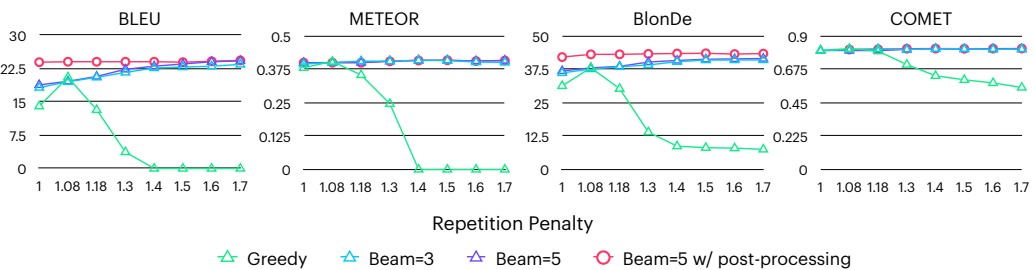

Figure 5: Automatic metric results across different decoding strategies. Repetition penalty $\gamma = 1$ represents pure greedy or beam search w/o penalty; $\gamma > 1$ denotes near-greedy decoding.

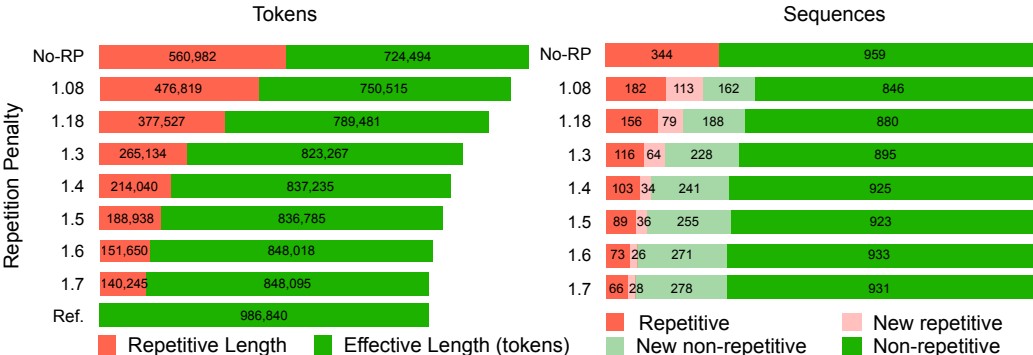

Figure 6: Left: **Effective token counts**; Right: **Sequence repetition analysis**. *(Non-)Repetitive* means sequences that staying (non-)repetitive w/ different $\gamma$; *New (non-)repetitive* indicates sequences that newly become (non-)repetitive. *No-RP* denotes no repetition penalty ($\gamma = 1$). *Ref.* means total number of tokens in the reference.

**Comparison of Decoding Strategies** By default, beam search is employed for all models, with beam size 5. However, upon training certain LLMs on the CH2CH task, we observe sub-optimal performance with beam search. We investigate the performance of three decoding strategy: *greedy*, *beam search* and *near greedy* decoding, which introduces repetition penalty $\gamma$ to discount the scores of previously generated tokens (Keskar et al., 2019).

Figure 5 presents the effect of applying the penalty $\gamma$ to both greedy and beam search decoding with different beam sizes. For beam search (with beam size = 3 or 5), both BLEU and BlonDe scores improve significantly. Concretely, with beam size = 5, BLEU and BlonDe increase from 18.80 to 24.20 and from 36.90 to 41.42, respectively. In contrast, the improvements in METEOR and COMET scores are comparatively smaller, suggesting that the overall translation quality may not be improving as expected. In addition, for beam search decoding, increasing $\gamma$ keeps improving translation performance and there are marginal variances across all evaluation metrics once $\gamma \geq 1.5$. For greedy decoding, however, translation quality rapidly declines when $\gamma > 1.2$.

We then explore the number of effective (i.e., non-repetitive) tokens generated as $\gamma$ increases ( Figure 6 (left)). We further analyze repetition sentence by sentence by separating test sequences into four categories: *repetitive*, *non-repetitive*, *new repetitive*, and *new non-repetitive* to illustrate how different repetition penalties would fare on the occurrence of repetition (Figure 6 (right)). In general, less sequences become repetitive as the penalty becomes stronger.

**Post-processing** To further evaluate the model's translation ability, we implement post-processing to eliminate repetitions in the generations. Before evaluation, we employ a sliding window with a length of 10 words, calculating the hash value of the substring within the window. As we slide the window, if the hash value of the current substring matches any previously seen hash value, we compare the actual substrings to confirm the repetition and then trim accordingly[5]. After cleaning, the blocks belonging to the same chapter are merged back together for evaluation at the chapter level.

---

[5]Most repetitions exhibit a self-reinforcement effect, continuously repeating the same sentences or phrases. Therefore, once a repetition is detected, we remove all subsequent words.

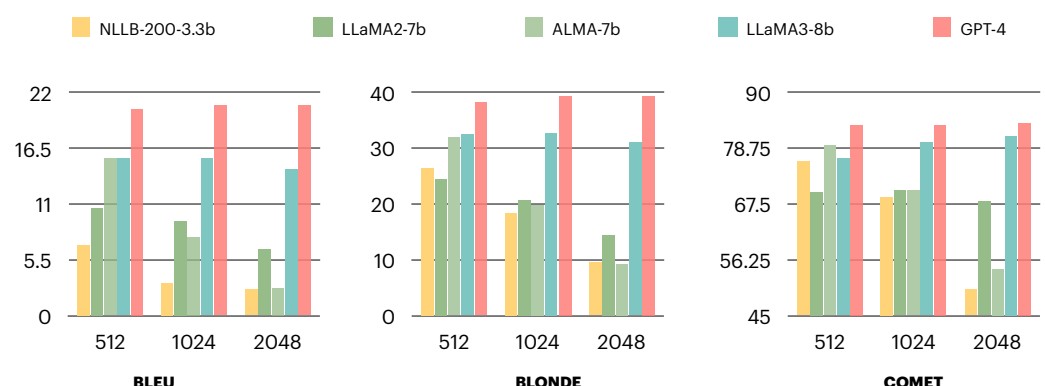

Figure 7: Zero-shot performance on JAM data across LLMs. The chapter-level data are segmented into chunks containing at most 512, 1024, 2048 tokens. ACL = average chapter length in tokens; The ACL of sampled instances=1850.

According to Figure 5, although applying repetition penalty in decoding procedure shows significant improvements in BLEU and BlonDe scores, the METEOR and COMET scores do not reflect similar gains. To determine whether repetition penalty genuinely improves translation quality rather than simply reducing repetition, we carefully examine the BLEU scores across the four categories before and after post-processing ($\rightarrow$). The division of the four groups is based on the results of $\gamma = 1.7$ compared to the case with no repetition penalty applied ($\gamma = 1$).

| $\gamma$ | Rep. | New rep. | New non-rep. | Non-rep. |
|---|---|---|---|---|
| 1.0 | $7.5 \rightarrow 9.4$ | $\mathbf{18.2 \rightarrow 18.2}$ | $8.3 \rightarrow 17.1$ | $\mathbf{22.5}$ |
| 1.1 | $8.4 \rightarrow 9.7$ | $10.0 \rightarrow 15.7$ | $12.5 \rightarrow 18.7$ | $\mathbf{22.5}$ |
| 1.2 | $8.9 \rightarrow 11.9$ | $11.0 \rightarrow 14.6$ | $13.6 \rightarrow 19.0$ | $22.3$ |
| 1.3 | $\mathbf{11.1 \rightarrow 13.0}$ | $11.8 \rightarrow 16.7$ | $16.0 \rightarrow 19.7$ | $22.3$ |
| 1.4 | $9.7 \rightarrow 12.5$ | $13.2 \rightarrow 16.4$ | $17.5 \rightarrow 20.2$ | $22.3$ |
| 1.5 | $10.0 \rightarrow 10.5$ | $13.5 \rightarrow 16.8$ | $18.9 \rightarrow 20.4$ | $22.2$ |
| 1.6 | $10.6 \rightarrow 11.4$ | $11.3 \rightarrow 15.0$ | $19.5 \rightarrow 20.4$ | $22.1$ |
| 1.7 | $7.8 \rightarrow 9.4$ | $5.9 \rightarrow 12.0$ | $\mathbf{20.7 \rightarrow 20.7}$ | $21.2$ |

Table 4: BLEU scores across different groups. $\rightarrow$ denotes after post-processing.

As Table 4 shows, the repetition penalty affects the four groups differently: for sequences that cease to be repetitive after the penalty is applied (*New Non-repetitive*), increasing $\gamma$ consistently improves translation quality. In contrast, for *Non-repetitive* sequences which stay non-repetitive before and after applying the penalty, increasing $\gamma$ slightly diminishes performance. It demonstrates that repetition penalty did not produce more meaningful translations for this group. On the other hand, applying an appropriate repetition penalty can slightly improve translation effectiveness for sequences that stay repetitive before and after applying the penalty (*Repetitive*). It should be noted that an excessively high penalty may negatively impact performance for sequences that are prone to repeat. Unsurprisingly, for sequences in *New Repetitive* which start to be repetitive after applying the penalty, the translation quality declines rapidly. This leads to a potential direction of future work to develop advanced decoding algorithms to avoid repetitions in translation.

## 5.3 How Do Large Language Models Perform on Literary Translation?

In order to evaluate the capacity of LLMs on CH2CH translation, we perform zero-shot evaluation on the JAM dataset across different models. To further analyze performance variations across different context lengths, we segment chapters into at most 512, 1024, and 2048 tokens, respectively. The results are presented in Figure 7.

GPT-4 outperforms all other models across both sentence-level and document-level metrics. Rather, translation-oriented models, such as NLLB-3.3B and ALMA-7B-Stage2, struggle in the CH2CH task, i.e., performance drop dramatically especially when the sequence become longer than 1024 tokens. One reason as to why ALMA-7B-Stage2 faces challenges in translating long sentences is that it has been finetuned exclusively on short parallel sequences. This may impair its capability to handle long-sequence translation and fully exploit the advantages of chapter-level contextual information to improve translation quality. However, we observe notable improvements after fine-tuning ALMA-7B on our chapter-level dataset JAM even in the most challenging setting where the context extends up to 2048 tokens, as shown in Table 3.

Despite LLMs such as LLAMA2 being theoretically capable of handling contexts of up to 4096 tokens, their performance in translation tasks over extensive contexts remains subpar. Before delving into more nuanced improvements in discourse-level translation, it is crucial to enhance the model's capacity for high-quality long-context translation.

**CH2CH vs. Sentence Translation**   The high-level objective of CH2CH translation is to leverage more training signals from chapter-level dataset. To test the effectiveness of this setting, we conduct an experiment to segment chapters into sentences for comparison. Concretely, we first split each chapter into separated sentences using the NLTK [6] package, then execute translation individually on each sentence with ALMA-7B. The translated sentences are concatenated back to calculate document-level evaluation metrics. Figure 8 indicates that ALMA-7B under the 512-tokens setting outperforms the sentence-segmented setting across all metrics, attesting the significance of CH2CH translation.

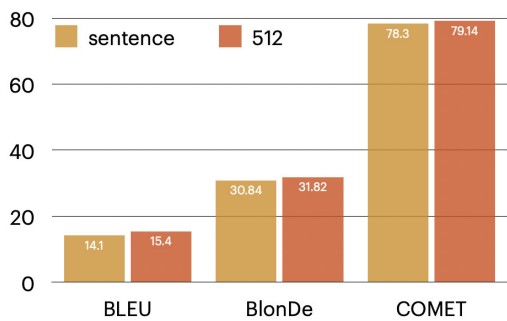

Figure 8: Zero-shot performance of sentence and 512-token segmentation.

**Decoder-only vs. Encoder-Decoder Architecture**   Under the zero-shot setting (Figure 7), ALMA-7B-Stage2 continues to surpass encoder-decoder translation model NLLB-200-3.3B on BLEU scores. In terms of document-level evaluation metrics, ALMA-7B-Stage2 performs on par with, or even better than NLLB-200-3.3B on the most BlonDe metrics, e.g., pronoun and discourse marker(d.m.). One potential explanation is that the backbone LLM LLAMA2-7B has a better context understanding and text generating ability. For example, discourse markers, e.g., *however, on the other hand*, are crucial for maintaining the coherence and cohesion of text, areas in which LLMs are trained. Furthermore, NLLB-200-3.3B tends to generate shorter text compared to other models. One hypothesis is that it is primarily trained on a sentence-aligned dataset, where the source and target sentences do not differ significantly in length.

After finetuning on JAM, though Encoder-Decoder perform slightly better than Decoder-only model, yet still under-perform ALMA models on most of the evaluation metrics (Table 3). The above results demonstrates the effectiveness of decoder-only models in handling complex literary translation. Particularly noteworthy is the fact that LLMs do not rely heavily on large amounts of parallel data and are inherently capable of translating long context sequences after finetuning.

## 6   CONCLUSION

While machine translation demonstrates strong sentence-level performance, it still falls short of human translation in effectively utilizing long-context information. In our paper, we show that Chapter-to-Chapter (CH2CH) translation is a viable approach for *context-aware* NMT, exemplified by our novel dataset, JAM. Chapter-level data, derived from professional translations, offers richer context signals and presents a more realistic scenario. Through detailed empirical experiments, we discover that LLMs are aptly suited for CH2CH translation following a two-step fine-tuning process: first at the sentence level, then at the chapter level. This procedure equips LLMs with a robust understanding of context, resulting in translations that are both coherent and context-aware. Nevertheless, challenges arise at the chapter level, notably the issue of repetition inheriting from LLMs' long-context generation, signaling the need for improved long-sequence decoding strategies in future research.

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

# APPENDIX: TOWARDS CHAPTER-TO-CHAPTER CONTEXT-AWARE LITERARY TRANSLATION VIA LARGE LANGUAGE MODELS

## A  JAM DATASET

### A.1  CORPUS INFORMATION

| Title | Author | Year | #Chapts | ACL (en/zh) |
|---|---|---|---|---|
| 1984 | George Orwell | 1949 | 24 | 5.8K/10.2K |
| A Tale of Two Cities | Charles Dickens | 1859 | 44 | 4.3K/8.0K |
| Alice's Adventures in Wonderland | Lewis Carroll | 1865 | 9 | 3.1K/5.7K |
| Ancient Greek Myths | / | / | 58 | 488.2/862.1 |
| Around the World In Eighty Days | Jules Verne | 1872 | 36 | 2.6K/5.5K |
| Black Beauty | Anna Sewell | 1877 | 13 | 1.9K/3.0K |
| Don Quixote | Miguel de Cervantes | 1605 | 125 | 4.4K/6.9K |
| Five Weeks in a Balloon | Jules Verne | 1863 | 44 | 3.1K/5.9K |
| How The Steel Was Tempered | Nikolai Ostrovsky | 1934 | 18 | 11.7K/24.8K |
| Little Prince | Antoine de Saint-Exupéry | 1943 | 28 | 822.3/1.4K |
| Little Women | Louisa May Alcott | 1868 | 47 | 5.8K/10.7K |
| Oliver Twist | Charles Dickens | 1838 | 53 | 4.4K/8.7K |
| Robinson Crusoe | Daniel Defoe | 1719 | 8 | 20.9K/35.4K |
| Tess of the d'Urbervilles | Thomas Hardy | 1891 | 59 | 3.7K/7.8K |
| The Adventures of Tom Sawyer | Mark Twain | 1876 | 35 | 3.1K/5.7K |
| The Moon and Sixpence | William Somerset Maugham | 1919 | 58 | 1.8K/3.9K |
| The Mysterious Island | Jules Verne | 1875 | 62 | 4.5K/8.2K |
| The Time Machine | H. G. Wells | 1895 | 13 | 3.4K/6.2K |
| Women in Love | D. H. Lawrence | 1920 | 27 | 10.3K/9.5K |
| Wuthering Heights | Emily Brontë | 1847 | 34 | 5.1K/9.3K |

Table 5: Corpus information for 20 sample books. ACL = average chapter length in tokens.

The whole JAM corpus contains world literatures; for a source text to be included in JAM, it must be (1) a literary work that has a published electronic version with chapter breaks along with (2) its corresponding human-written, Chinese translations from professional translators available on the Internet. Books genres include both fiction (e.g., romance, science, adventure, etc) and non-fiction literature (e.g., biography and self-help).

All books in JAM have entered the public domain with cleared copyright, from the earliest published in 1817 to the latest in 1949. Table 5 shows 20 sample books from the JAM dataset, in which the ACL column is obtained by using LlamaTokenizerFast.

## B  IMPLEMENTATION DETAILS

### B.1  DATA

Data for baseline models is encoded and vectorized with byte-pair encoding Sennrich et al. (2016) using the SentencePiece (Kudo & Richardson, 2018) framework. We use a 32K joint vocabulary size for Zh→En. Full corpus statistics of WMT22 are in Table 6.

| Dataset | Lg. Pair | Train | Valid | Test |
|---|---|---|---|---|
| WMT22 | Zh→En | 25134743 | 2002 | 2001 |

Table 6: Sentence counts across WMT22 datasets.

To segment JAM chapter-level dataset into chunks, we first decide the number of chunks to split in a chapter by ensuring that each chunk includes no more than 2048 English and Chinese tokens, then equally segment the chapter into the computed number of chunks. There is no overlap between chunks, and we keep a sentence a complete unit when we split chapters.

| Model | BLEU | BlonDe | | | | | COMET | ACL |
|---|---|---|---|---|---|---|---|---|
| | | all | pron. | entity | tense | d.m. | | |
| *512 tokens* | | | | | | | | |
| NLLB-200-3.3b | 6.90 | 26.37 | 63.26 | 23.96 | 63.53 | 61.59 | 0.7592 | 870 |
| LLaMA2-7b | 10.60 | 24.49 | 73.89 | 17.51 | 72.70 | 66.85 | 0.6990 | 1551 |
| ALMA-7b | 15.40 | 31.82 | 88.35 | 19.69 | 88.22 | 82.30 | 0.7914 | 1608 |
| GPT-4 | **20.40** | **38.24** | **91.03** | **39.43** | **90.34** | **82.35** | **0.8324** | **1863** |
| *1024 tokens* | | | | | | | | |
| NLLB-200-3.3b | 3.20 | 18.32 | 47.37 | 17.17 | 46.15 | 44.29 | 0.6888 | 709 |
| LLaMA2-7b | 9.30 | 20.57 | 64.09 | 11.60 | 66.44 | 59.74 | 0.7025 | 1648 |
| ALMA-7b | 7.70 | 19.82 | 68.49 | 13.30 | 71.00 | 62.49 | 0.7017 | 2223 |
| GPT-4 | **20.60** | **39.20** | **91.12** | **40.87** | **90.32** | **82.87** | **0.8347** | **1821** |
| *2048 tokens* | | | | | | | | |
| NLLB-200-3.3b | 2.50 | 9.48 | 41.62 | 7.37 | 50.66 | 25.98 | 0.5009 | 1254 |
| LLaMA2-7b | 6.40 | 14.40 | 49.45 | 8.63 | 53.66 | 39.69 | 0.6778 | 1780 |
| ALMA-7b | 2.70 | 9.09 | 42.27 | 6.35 | 47.98 | 27.77 | 0.5433 | 2382 |
| GPT-4 | **20.70** | **39.35** | **91.39** | **41.81** | **91.39** | **83.67** | **0.8359** | **1765** |

Table 7: Zero-shot performance on JAM data across LLMs. The chapter-level data are segmented into chunks containing at most 512, 1024, 2048 tokens. ACL = average chapter length in tokens; The ACL of sampled instances=1850.

## B.2 BASELINE TRANING

We train baseline models (Encoder-decoder and Decoder-only) on the `fairseq` framework . Following Vaswani et al. (2017); Fernandes et al. (2021), we use the Adam optimizer with $\beta_1 = 0.9$ and $\beta_2 = 0.98$, dropout set to 0.3, an inverse square root learning rate scheduler with an initial value of $10^{-4}$, and the warm-up step set to 4000. Here, we only train the Transformer `base` version, and the decoder-only model is also derived from the base Transformer `base` architecture. We keep the parameter size of both Encoder-decoder and Decoder-only architecture similar for fair comparison.

## B.3 LLM TRAINING

All models are trained with 8xA40 GPUs and DeepSpeed+ZeRO3. Following Xu et al. (2023a), we use Adam optimizer, weight decay set to 0.01, and the warm-upratio set to 0.01, an inverse square root learning rate scheduler with an initial value of $2 \times 10^{-5}$.

The zero-shot evaluation on JAM dataset across different chunk sizes are shown in Table 7.

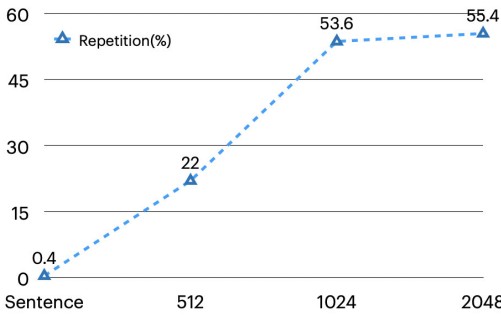

Figure 9: Repetition ratio in the generation results for different input context length

## B.4 REPETITION ANALYSIS ON ZERO-SHOT TRANSLATIONS

As illustrated in Figure 9, repetition is not an issue for sentence-level translation. However, the repetition ratio significantly increases as the input context length increases from 512 to 1024. Furthermore, Figure 10 shows that as the input length increases, the repetition start position also occurs earlier.

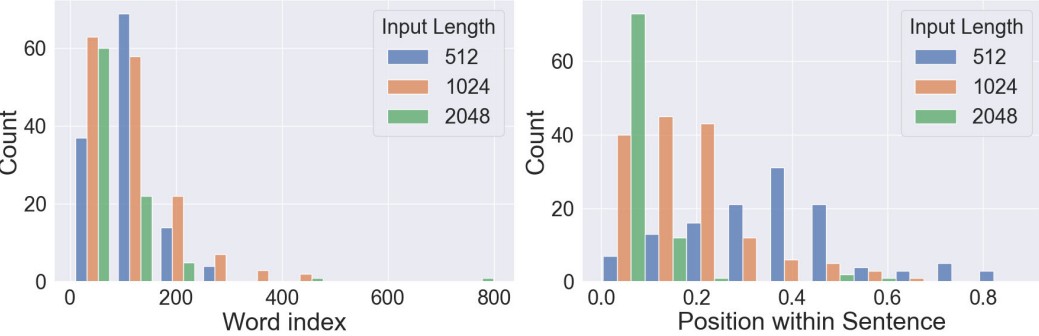

Figure 10: Repetition start position across different input lengths. Left: The word index of repetition, Right: The relative position of repetition.

## B.5 POST-PROCESSING ON FINE-TUNE TRANSLATIONS

Post-processing eliminate repeated words and phrases in generated translations. Table 8 shows a comprehensive automatic metric comparison between translations with post-processing versus. without post-processing.

| Model | WMT | JAM | Post-processing | BLEU | BlonDe | | | | | COMET |
|---|---|---|---|---|---|---|---|---|---|---|
| | | | | | all | pron. | entity | tense | d.m. | |
| ALMA-7B-Stage1 | ✗ | ✓ | ✗ | 15.70 | 33.46 | 74.28 | 30.62 | 70.11 | 71.72 | 0.7806 |
| ALMA-7B-Stage2 | ✓ | ✓ | ✗ | 18.80 | 36.90 | 81.34 | 32.72 | 77.83 | 76.81 | 0.8025 |
| ALMA-7B-Stage1 | ✗ | ✓ | ✓ | 21.6 | 39.54 | 86.43 | 35.43 | 84.52 | 82.98 | 0.7986 |
| ALMA-7B-Stage2 | ✓ | ✓ | ✓ | **23.9** | **42.73** | **90.69** | **38.41** | **89.02** | **84.95** | **0.8106** |

Table 8: Automatic metric result of ALMA-7B translations on JAM, with versus without post repetition removal processing. **Bold** denotes best performance.

