# OpenReview forum: "Towards Chapter-to-Chapter Literary Translation Via Large Language Models"
_ICLR.cc/2025/Conference — ICLR 2025 Conference Withdrawn Submission_

### Official Review · Reviewer_ZB2k · 2024-10-29

**Soundness:** 2
**Presentation:** 3
**Contribution:** 2
**Rating:** 3
**Confidence:** 4

**Summary:**

This paper presents a new dataset for "chapter-to-chapter" machine translation, called JAM. The JAM dataset contains 4.2K parallel chapter pairs from 132 books in Chinese-English literature, and on average each en/zh pair contains ~2K words.

The main difference from prior work in document-level MT such as Thai et al. 2022 [1], the dataset does not assume a 1:1 mapping between paragraphs in the source and target language. Instead, the mapping is at a chapter level (so the source/target need not have the same number of paragraphs).

The authors conduct a number of experiments on their newly introduced dataset, including (1) seq2seq models trained from scratch; (2) models pretrained on WMT sentence level translation data before JAM fine-tuning; (3) zero-shot LLMs (like GPT4) for translation. Overall, the authors conclude that the most effective small sized models are obtained after a two stage fine-tuning: sentence level data followed by JAM. The authors also find GPT-4 is surprisingly effective, seemingly outperfoming fine-tuned models.

[1] - https://aclanthology.org/2022.emnlp-main.672/

**Strengths:**

1. This paper studies an important and relatively underexplored topic: document-level machine translation. Moreover, the contributions are in the literary domain which has important downstream applications: translations of novels / articles.

2. The paper contributes a nice document-level MT dataset of 4.2K parallel en/zh pairs at a chapter level.

3. The paper includes a number of interesting ablations: enc-dec vs decoder-only, comparison of decoding strategies, slicing by token repetition amounts, chapter-level vs sentence-level translation.

**Weaknesses:**

1. **The paper is lacking a human evaluation**, and it's unclear if BLEU / Blonde are suitable to evaluate JAM outputs. [1, 2, 3] suggests that BLEU / Blonde are very poor proxies for evaluating document level machine translation. On a related note, Figure 8 shows a very small difference between a chapter-level and sentence-level translation. This further makes me think the metric maybe not capturing the unique aspects of document-level MT. I suggest the authors to use aspects of the human evaluation from [4] to complement their automatic metrics.

2. **The technical contribution in the paper feel fairly incremental compared to Par3** ([1]). Both Par3 and JAM operate in a literary domain, and Par3 contains many more language pairs beyond en/zh. The chapter level mapping of JAM is a nice addition, but it's not empirically clear how much this adds over a paragraph-level aligned dataset.

3. **The paper has a limited focus on zero-shot pretrained LLMs baselines**, which according to Figure 7 / Table 3 seem to be the strongest at this task and are being preferred for translation applications. There are a number of interesting questions to explore here: (1) do specialized long context models like Gemini 1.5 Pro do better on JAM? (2) Are Chinese-focused LLMs better than English-focused LLMs? (3) Do pretrained LLMs use a 1:1 mapping at a paragraph/sentence level, or do they perform discourse-level operations? (4) Does GPT-4o / Gemini 1.5 Pro solve some of the critical translation errors listed in [4]?

4. **No qualitative examples or analysis was provided** for the dataset or for the translation model outputs. These are critical to get a general feel of the quality of the dataset, and understand the weaknesses in document-level MT for current models.

[1] - https://aclanthology.org/2022.emnlp-main.672
[2] - https://statmt.org/wmt22/pdf/2022.wmt-1.6.pdf
[3] - https://arxiv.org/pdf/2302.09210
[4] - https://arxiv.org/pdf/2304.03245

**Questions:**

Which GPT-4 variant is used? Did the authors try GPT-4o or GPT-4o-mini?

Why was segmentation of chapters needed for the GPT-4 experiments? A 2048 token segmentation will cut out 90% of the chapter for longer chapters like Robinson Crusoe.

Please combine Table 3 with Figure 7, so that it's possible to direclty compare fine-tuned models with zero-shot models like GPT-4.

Nit: It would be more helpful to provide the average number of sentences / words *per* en/zh article in Table 2.

---

### Official Review · Reviewer_H8s9 · 2024-11-03

**Soundness:** 3
**Presentation:** 4
**Contribution:** 3
**Rating:** 6
**Confidence:** 4

**Summary:**

The paper introduces a novel approach to context-aware machine translation by proposing a chapter-to-chapter (CH2CH) translation setting. The authors curate a new Chinese-English literary dataset, JAM, consisting of 132 books to facilitate research in this area. They evaluate the performance of common machine translation models within the CH2CH framework and propose a fine-tuning procedure for large language models (LLMs) to improve translation quality. Furthermore, the paper highlights the challenges associated with chapter-level translation through comprehensive analysis.

**Strengths:**

- The paper introduces a new translation setting that shifts focus from sentence and paragraph alignments to chapter-level, marking a significant advancement in the field of machine translation.
- It presents a new Chinese-English literary dataset, JAM, consisting of 132 books to support research in this area.
- The paper uncovers challenges in chapter-level translation through detailed analysis.

**Weaknesses:**

- The paper could benefit from a more detailed comparison with existing translation models and commercial translation systems, such as Google Translate and Bing Translator.
- While the JAM dataset is a valuable resource, the authors might consider discussing the potential biases or limitations inherent in a dataset derived from literary works.

**Questions:**

- Could you discuss any potential biases present in the JAM dataset, and do you plan to extend the CH2CH translation approach to other domains?
- Are there plans to test the proposed fine-tuning procedure on languages other than Chinese and English? What challenges might arise in such efforts?
- The paper addresses the issue of repetition in LLMs. Could the authors suggest potential solutions or improvements to decoding algorithms to mitigate this problem?
- Can the authors elaborate on the specific challenges posed by long-context translation and how their proposed methods address these challenges?

---

### Official Review · Reviewer_zh7K · 2024-11-05

**Soundness:** 2
**Presentation:** 2
**Contribution:** 2
**Rating:** 3
**Confidence:** 5

**Summary:**

The paper proposes a Chapter-to-Chapter (CH2CH) translation task, which is a more practical and challenging translation scenario. To support this task, the authors constructed a dataset of 132 literary works along with their professional Chinese translations and investigated the performance of common machine translation models under the CH2CH setting.

**Strengths:**

1. The paper introduces a Chapter-to-Chapter translation dataset, JAM, which provides new resources and challenges for document-level translation research.
2. Through experiments, it offers a thorough analysis of the main challenges encountered in CH2CH translation.

**Weaknesses:**

Lack of innovation. The paper employs repetition penalties and the removal of redundant clauses to address translation repetition issues, which is conventional. Overall, the paper lacks substantial methodological contributions.

**Questions:**

See Weakness.

---

### Note · Authors · 2024-11-25

I have read and agree with the venue's withdrawal policy on behalf of myself and my co-authors.